# Coalition Formation among the Cooperative Agents for Efficient Energy Consumption

**Areej A. Malibari** [1] , **Daniyal Alghazzawi** [1,*] and **Maha M. A. Lashin** [2,3]

1   Faculty of Computing and Information Technology, King Abdulaziz University, Jeddah P.O.Box 80200, Saudi Arabia; aamalibari1@kau.edu.sa
2   College of Engineering, Princess Nourah bint Abdulrahman University, Riyadh P.O.Box 11671, Saudi Arabia; mmlashin@pnu.edu.sa
3   Mechanical Engineering Department, Faculty of Engineering Shoubra, Banha University, Cairo 13511, Egypt
*   Correspondence: dghazzawi@kau.edu.sa

**Abstract:** Energy saving is a significant research area in Saudi Arabia; however, significant problems have emerged related to its distribution and consumption. Use of an agent is assumed to combat these problems by forming efficient coalitions to control the energy consumption and energy distribution process. This study presents a novel algorithm for distributing the value calculation among the cooperative agents. This is likely to reduce the consumption of energy and extend the coalition lifetime used. The developed algorithm is compared with the basic modified coalition formation algorithm for evaluating its effectiveness. The results showed a reduction in cooling consumption by 20% after applying optimization algorithms. The amount of reduction in the cooling consumption reflects a 31% reduction in expected cooling costs, without affecting the household comfort. Therefore, the study concludes that DNsys provided better performance than the NNsys.

**Keywords:** agent; coalitions; consumption; DNsys; DCVC algorithm; cooling consumption; energy distribution

## 1. Introduction

This paper works on a commercial scenario, in which the UAV's can belong to different vendors, and they can gather some type of monetary advantages for participation in each mission. Both the efficiency and task performance were accounted for, undertaking the resource constraints throughout the coalition formation from the perspective of leaders, as well as the individual preferences throughout the decision-making of followers for joining the available coalitions in the proposed coalition formation model. The UAVs might portray selfish behaviors by not using the resources that they originally committed to throughout the coalition formation in a realistic commercial setting, with the benefit of protecting these resources for future missions in obtaining advantages. A novel reputation-based mechanism has been developed in this proposed model for keeping the record of the cooperative behaviors of UAV.

Agent, a computer system, operates anonymously in an environment to meet its delegated objectives [1]. An intelligent agent is the one that executes timely and flexible actions, constituting reactive, proactive, and social behaviors [2]. Intelligent agents can interact with other agents or humans through cooperation, coordination, and negotiation [3]. Figure 1 demonstrates the agent lifecycle, which requires a comparison with the information of the previous agent for the environment information (perception), following determining the action to be implemented (decision), and lastly, executing the decision (action) [4].

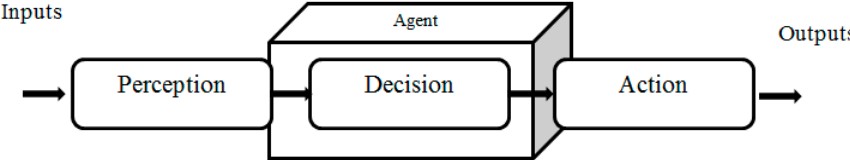

**Figure 1.** Agent Life Cycle.

Several agents are present in a multi-agent system (MASs) (Figure 2), and interact by exchanging messages using a computer network infrastructure [5]. A multi-agent system can be centralized (one central agent undertakes the collection of partial plans from agents) or decentralized (central agent does not control agents) [6]. The decentralized policies must assume merely partial system knowledge in each agent and must tackle communication comprehensively, whereas the centralized policies indicate the decision of the agents based on the global system state.

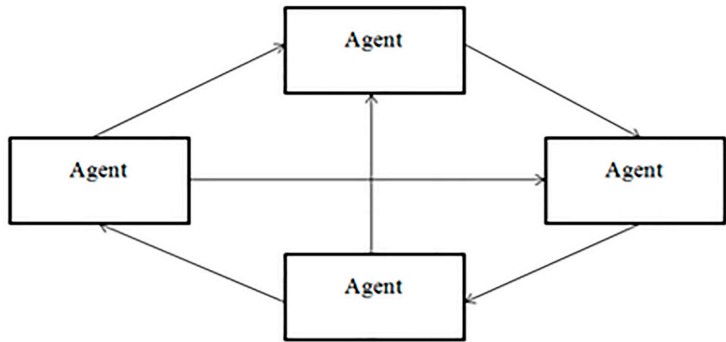

**Figure 2.** Simple Multi-Agent System.

The agents need to coordinate with each other for devising a relief plan, which enables them to reach a consensus [4]. In networking, negotiation refers to the process of cooperation among agents. Coalition formation (CF) allows agents to negotiate and coordinate to minimize the aggregate demand of the local electricity grid [7]. It can be applied through two methodologies;

Cooperative game theory [8], which requires the safety of a coalitional to block sub coalitions or different gatherings to meet the coalition being referred to;

Non-cooperative collating [9], in which people make recommendations to form a coalition that can be acknowledged or dismissed.

The main four areas of energy consumption are heating, cooling, water heating, and lighting [10]. Most of the consumption comes from space heating and cooling, such as air conditioners. However, the main problem is how energy is distributed and how resources are consumed, rather than how energy is used. Taleb and Sharples [11] stated that agents could regulate and minimize energy consumption and distribution by forming efficient coalitions. There is a need to develop an efficient algorithm for distributing the coalition value among multiple agents by reducing energy consumption for more than one building in the same area. There is a need to compare the efficiency of the algorithm with some common coalition formation algorithms to evaluate its effectiveness [12].

Players in cooperative games work collectively and try their best to enhancing the overall benefit of the coalition as compared to the uncooperative method. Recently, coalition game theory has been implemented to model WSNs or resource allocation. Different approaches were provided for MAC address assignment, transmission scheme, routing protocol, and sleep time allocation. However, the modeling and formation of cooperative collation in these methods have not been concerned with prolonging the lifetime of WSN. Previously, the data transfer model was used for WSNs with the implementation of the Shapley value and Nash equilibrium; however, the formation of coalitions has not been

understood, as well. The energy-efficient coalition game model was proposed for WSNs, however, the data transmission strategies have not been undertaken.

This paper proposes a new distributed compressive sensing-based data collection approach, using spatial-temporal association for enhancing the network performance with respect to data reconstruction accuracy and energy consumption. A spatial correlation-based coalition development algorithm has been proposed for localizing data gathering of the sensor nodes. This tactic exploits the sparsity distribution of signals for correlating group nodes spatially into coalitions. The sparsity distribution of signals defines a utility function. This function is utilized for reducing the number of active sensor nodes and mitigating energy consumption. A spatial temporal correlation-based compressive sensing solution has been proposed within each coalition after coalition formation. A block diagonal measurement matrix has been employed in the proposed solution for producing a linear combination of sensor node readings.

## 2. Related Work

Energy efficiency has been considered in the literature with respect to cooperative communications. The power consumption encompasses the transmission power consumption, accounting for the energy efficiency of the peak-to-average power ratio and power amplifier, and the processing power consumption of the radio frequency aspects at the receive and transmit RF chains [13–15]. In a group, single-antenna nodes have been used with each other in transmission to another cluster of single-antenna nodes via distributed multiple-input, multiple-output techniques, including general space-time block coding or Alamouti's space-time block coding [16]. Additional training overheads were taken into account in the work of [17] for the purpose of analyzing the effects of the channel path loss exponent on the energy efficiency. In addition, a source code, in [18], was transmitted to a destination node with the help of N relay nodes.

Energy efficiency is not guaranteed at all times because of additional processing power consumption in retransmission and reception of information in cooperative communications. Therefore, previous studies have discussed the energy efficiency over transmission distance and witnessed the situationsfort which cooperative communications must be integrated [19–22]. Common gaps in the previous studies are that they merely undertake simplified network settings with the similar distances among network nodes. Additionally, they do not undertake any mechanism utilized for exchanging and attaining channel state information on the basis of computing transmission power consumption [23].

The gain and cost in cooperation are considered within coalition formation games. Nodes, when forming cooperative clusters, accomplish transmission power saving via spatial diversity for energy efficient cooperative communications where it induces additional processing power because of the retransmission and reception of eavesdropped information. Previous studies have also proposed merge and split rules for finding a better cooperation structure along with the application of and iterative merge-and-split process, while the authors have undertaken that channel state information was available at the nodes [24–26]. Information exchange has been actualized for obtaining particular medium access control when undertaking implementation feasibility. Additionally, the difficulty of exchanging information is too high for exchanging information in computation complexity and wireless networks specifically with large numbers of nodes. Thereby, a heuristic time-division multiple access was proposed on the basis of the merge process to form cooperative groups [27]. Three stages were included in the merge process, such as merge, cooperative transmission, and transmission request. For the individual members, the condition for a merge is that the merge drives to power saving for the group regardless of causing further power stress.

The application of a merge process is valid to a cooperative communication protocol, which includes wireless network co-cast because of its competence for overcoming the concerns of imperfect timing synchronization and frequency caused by the asynchronous cooperative nature [23]. Nonetheless, the proposed process can be integrated into other

protocols with anticipated minor changes in cooperative communications. The power consumption for individual nodes is characterized in direct transmission as well as in cooperative transmission [24]. The performance of the proposed process was then evaluated for comparing the performance between the iterative merge-and-split process and merge process.

The value of each coalition should be computed and the coalitions compared for finding the outcome of a characteristic function game. A characteristic function game has the property that each value of a coalition needs to be computed only once, as it undertakes each value of coalition as independent, static, and deterministic [25]. Shehory and Kraus (1995; 1996; 1998) have originally proposed this property for reducing the number of redundant computations. They instigated an algorithm where each agent negotiated over which coalition must be assigned to its value calculation share rather than each agent computing each coalition value in which it is a member [28]. On the contrary, several limitations affect this algorithm, such as that there is no guarantee that every coalition is computed once and merely once; a number of messages should be sent between the agents for facilitating the negotiation; and there is no guarantee that the number of coalition value computations is almost equally performed by each agent [29].

Chao et al. [30] have emphasized solutions for offering lasting battery lifetime and proposed enhanced mechanisms for radio access networks, core networks, and M2M devices for acquiring better power saving for M2M devices. It was observed that these M2M devices work efficiently with simplified activities under optimized signaling flow. Similarly, Estevez [31] has shown that Green IoT might enhance energy efficiency, and assist in reducing environmental pollution and monitoring greater portions of the environments. Greater energy savings are promised through energy efficient protocols and scheduling techniques. Energy independence is a reality because of energy harvesting and pollution control is becoming more pervasive and smarter. Wu et al. [32] have identified that the effects of sustainable development goals (SDGs) would be positive, significant, and profound on the activities of human beings. Information and Communications Technology (ICT) has lasting sustainability issues to be solved. On the contrary, ICT would have significant potential to play essential and fundamental roles for supporting global social, economic, and environmental sustainability. Lorincz et al. [33] have discussed different paradigms for wireless access networks, including device-to-device communications, massive multiple-input multiple-output communications, millimeter-wave communications, and long-term evolution in unlicensed spectrum for enhancing wireless networks energy efficiency. In addition, approaches associated with the resource management, power management, and green monitoring are promising approaches to enhance DC power usage efficiency.

Clustering is one effective method in WSN [34–38] to save energy and maximize network life. In the cluster system, WSN nodes are divided into clusters and the cluster header (CH) acts as a node to coordinate other members of each cluster. CH is responsible for collecting, compiling, and sending the data collected directly or by multi-hop to the base station (BS). The clustering system reduces the power consumption of the network by the following [39,40]: (1) reduce long-distance transmission; (2) minimize the functionality of each node; (3) reduce the transmission packet by collecting data in CH for transmission volume, thus saving the energy consumption of the network.

Cooperative theory involves three levels of cooperation: formation of an alliance (that is, who agrees to work together), team building (that is, who agrees to use what resources to do what tasks), coordinated cooperation (that is, reaching an agreement on how to work together), and strong coordination [40]. We have discussed in detail the basics of HCF in the literature [41], the P2P energy transfer with the game theory method in [42], the powerful structure of alliance building in [43], the alliance game to achieve effective group solutions in [44], the Collaborative Policy Alliance formation, and the general method of alliance formation. However, given the unreliability of energy and the limitations of municipalities, it is a challenge to find the best producers and consumers with a surplus of

energy for those who need energy, and this challenge has not been properly addressed in the literature.

This work solves the alliance formation model proposed by network operators to encourage small participants to participate in a flexible market, including the Shapley values in the proposed model, as well as the potential revenue that consumers receive when they sell their flexibility; one can also look at the fairness of the alliance to find the best alliance structure. An optimization model based on differential distribution is also proposed as a method of finding the best alliance structure based on a specification.

The block diagonal measurement matrix is organized appropriately in a way that balances the communication and computation load over the coalitions. This spatial-temporal association based compressive sensing is utilized within each coalition for compressing sensor node readings as well as transferring it to the base station. A joint sparse signal recovery mechanism has been applied in the base station when receiving compressed data. Initially, a joint sparse signal recovery is executed for finding the common sparsity profile among the coalitions. This recovery process is conducted within each coalition for achieving a common profile among sensor nodes. Better accuracy is achieved with the utilization of a recovery algorithm in data reconstruction where the number attained measurements are mitigated. On the contrary, this study has analyzed the effect of different transform bases on compressive sensing. Different transform bases represent the sparse or compressible signal. Selecting the adequate transform basis is important in sparse representation, leading to fewer measurements and additional signal reconstruction throughout the recovery phase.

### 3. Methodology

The study proceeds by providing an overview of the current building structure. Following that, descriptions are provided about the selected buildings (nodes). Distribution of energy used within the buildings is also provided, followed by suggested modifications and potential improvements derived from the modeling. The energy consumption within the buildings was analyzed using the Log Tag Analyzer and MATLAB software. Simulation was presented based on real weather data and time from the particular files. The model designed for urban unit is therefore generalized for facilitating the broader use, to develop urban design and thermal simulations. Conventionally, a visual assessment or an automated GIS platform was selected for visiting the site of interest. This will be sufficient for allowing the reference offsetting, relative humidity, and hourly air temperature for the chosen buildings for different seasonal climate forcing. A coalition is created exclusively for air conditioning, while the rest of the uses continue to be attended by the zone distributor.

The distribution of coalitional value calculations (DCVC) algorithm is a coalition formation technique, which guarantees that every single agent receives a portion of the coalitional value calculation. The non-negotiable system is appropriate to resolve the issues, as it offers agents more control as compared to other protocols that are centralized and need a mediator whose part is to share information between agents. The share of the agent calculations is considered both exhaustive and disjoined [45]. The distributive negotiation system (DCVC algorithm) attempts to distribute a "fixed pie" of benefits that operates under zero-sum conditions and assumes one person's gain is another person's loss [46,47]. In a distributive negotiation, each side adopts an extreme or fixed position, despite knowing that it will not be accepted, and then seeks to cede at minimum before reaching an agreement. Prospect Theory indicates that distributive negotiation leads to a loss in peoples' value as compared to their gains. These are more risk-averse about losses; whereas, concession–convergence bargaining is likely to be more acrimonious and less productive of an agreement [48].

The case of Yanbu city, that is, a residential area, is selected. The city is located on the red sea coast of western Saudi Arabia. The rationale for selecting this study area is its diverse and rapid commercialization. It is useful to consider the climatic conditions affecting the area while analyzing building energy consumption. The climate in Yanbu

during the summer is characterized by fierce heat and high humidity, where the level of humidity reduces with warmth and occasional drizzling in November and December [11]. This study was carried out in 2013 at a residential area that contains ten houses, one mosque, and one hospital. Each of the building structures was represented by a node in the structure (12 nodes) (Figure 3).

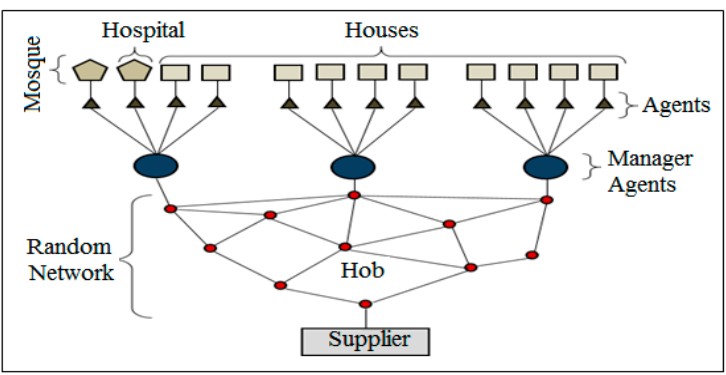

**Figure 3.** Structure of the Residential Area in the Case Study.

### 3.1. Modification of the DCVC Algorithm

The DCVC algorithm is modified to calculate different computational speeds. An equal amount of time is required to calculate their value share and the shared values of two agents [49]. It is inefficient to have each agent calculate the same number of values for agents to have significantly different computational speeds. The distribution is to be done based on the agents' relative computation speed. Dijkstra's algorithm [18] was used to indicate the shortest path between agents (Figure 4). There are a wide range of potential implications of coalition formation in multi-agent systems as a coalition of 12 nodes in the present algorithm can form in a network that reduces the transmission costs, power consumption, and increase network security. The distributed vehicle routing offers the formation of a coalition for reducing transportation costs by sharing deliveries. These coalitions are likely to receive optimal resource allocations from the grid, which is also known as grid computing. The DCVC algorithm proposed in this study distributes coalition value calculations as soon as the graph presents agents in the form of nodes on a graph, where an edge between them point towards some synergistic link.

---

Each agent $a_i$ should perform the following:

- Sort the set of agents based on the agents' UID in an ascending order.
- Set: $\alpha = 1$.
- For every $s \in S$, do the following:
  1. If $(N_s \geqslant n)$ then:
     1.1. Calculate the size of your share: $N_{s,i} = \lfloor N_s / n \rfloor$
     1.2. Calculate the index of the last coalition in your share: $index_{s,i} = i \times N_{s,i}$
     1.3. Calculate the values of each coalition in your share.
     1.4. Calculate the number of additional values that need to be calculated: $N' = N_s - (n \times N_{s,i})$
     Otherwise:
     1.5. Calculate the number of additional values that need to be calculated: $N' = N_s$
  2. If $(N' > 0)$ then:
     2.1 Find the sequence of agents $A'$ in which each agent should calculate one additional value, and if you are a member of $A'$, then calculate the appropriate value. This is done as follows:
        ○ If $(\alpha + N' - 1 \leqslant n)$ then: $A' = (a_\alpha, a_{\alpha+1}, \ldots, a_{\alpha+N'-1})$
              else: $A' = (a_\alpha, a_{\alpha+1}, \ldots, a_n, a_1, \ldots, a_{(\alpha+N'-1)-n})$
        ○ If $(a_i \in A')$ then calculate one of the additional values based on your position in $A'$
        ○ If $(\alpha + N' \leqslant n)$ then: $\alpha = \alpha + N'$, else: $\alpha = \alpha + N' - n$

**Figure 4.** DCVC Algorithm.

The structure considered in this study is comprised of 12 nodes, *n* equals, and each node has its own *ai* agent. The nodes' order is prioritized based on their p and don values. This process is repeated until all the coalitional values in the list are calculated. Figure 5A,B explains the steps of fair energy distribution. First, each values needs are calculated using a different agent. Following this, the central decision-maker negotiates between agents, which show the priority-based allocation.

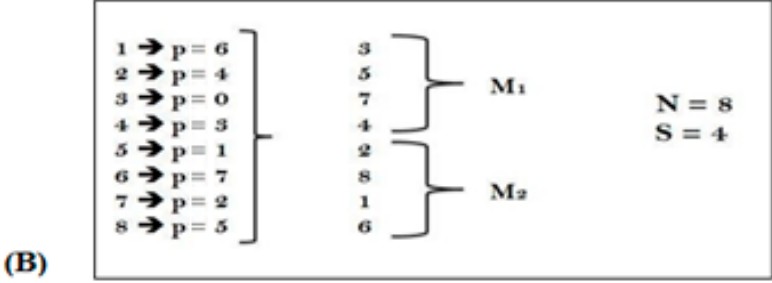

**Figure 5.** Priority-Based Allocation for all possible 8 Agents.

### 3.2. Coalitional Agent Selection

Deals with multi-agent CF problems propose priority-based allocation and the remaining rate feature [50]. A high agent means this agent is never shut down, nor does it leave the area of communication logically. The agent calculates the remaining rate according to its arrival time, current time, and the agent's average staying time in the environment. The rate becomes higher as the staying time is extended.

### 3.3. Complexity Analysis

The DCVC algorithm has a complexity far lower compared to the coalition formation issue in optimal manner which is NP-hard. Indeed, all the possible partitions were checked for finding the coalition formation issue, which is equal to the M-th Bell number (BM) for finding the optimal partition of a set of M players. It should be noted that the BM is acquired by the recursion:

$$B_{n+1} = \sum_{k=0}^{n} \binom{k}{n} B_n, B_0 = B_1 = 1 \tag{1}$$

The complexity of the proposed DCVC algorithm relies on the number of investigations performed in every iteration or node, which relies on the number of nodes in the network. Indeed, each coalition requires investigation of the merge with all the other coalitions in $\Pi$. Therefore, the total number of merge attempts is approximately $O(|\Pi|)^2$, which relies on the number of coalitions and not on the number of nodes in the network. On the contrary, the complexity of the DCVC operation for each coalition is $O(|\Pi|)$.

After sorting the coalition lists, all manager agents took their places and the energy required to supply all the operating cooling devices in their share was calculated [50].

The result of the calculation was provided to the supplier before starting the energy distribution (Figure 6).

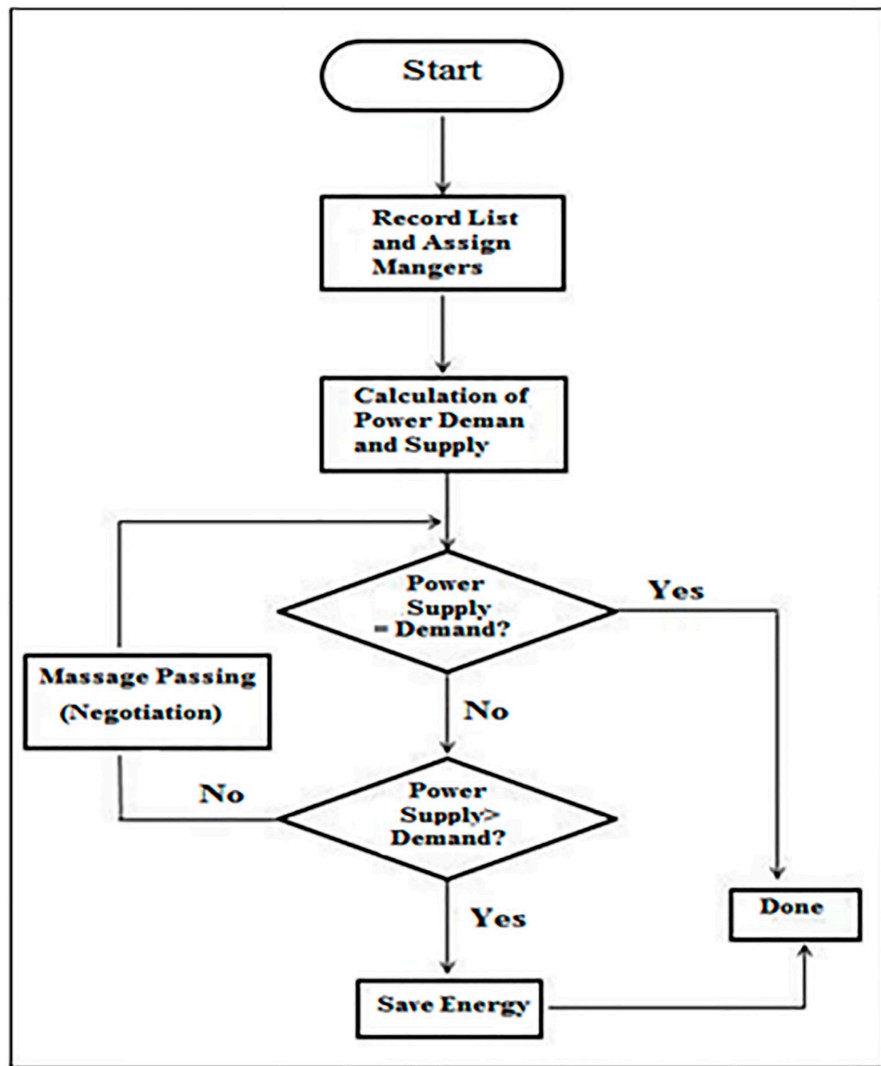

**Figure 6.** Distribution Process after Modification.

The total number of attempts in the Π induces finding all likely partitions of the coalition, offering a Π complexity for the coalition. Therefore, the complexity of the split operation is closely associated to the size of the coalition and not on the total number of users in the system. On the contrary, only CH are in charge of executing the heuristic algorithm in the proposed scheme in order to determine the appropriate combined packets. Therefore, the complexity of checking the connectivity of each vertex with the other vertices and renewing its subsequent weight and layer is confined to the cluster size.

### 3.4. Implementation Tool

MATLAB software was used to implement the proposed algorithm. The DCVC algorithm was modified to perform calculations at different computational speeds. A new mechanism to address multi-agent coalition formation problems based on priority-based allocation and the remaining rate feature was proposed and described.

### 3.5. Data Collection

The Royal Commission for Yanbu, Saudi Arabia, was selected because the styles and cooling systems of the houses in the area were the same. Ten houses, one mosque, and

one hospital were selected. The data collected included observations of nodes' (houses, mosque, and hospital) internal temperatures, prayer times, and the time of the moment.

Energy loggers were placed in each home to read the internal temperatures during the day. The loggers were set to read the current temperature every single minute for three periods, which were five days long each, in the summer of 2013. The periods were 5–9 June, 15–19 June, and 29 June–3 July. The prayer times were collected from a file containing the annual prayer times. The moment time was collected from the clock on the device on which the system was working in.

## 4. Proposed Distributed-Negotiated System (DNsys) Description

The DNsys code's flow chart contains many phases; the first one contains start and initialization of network random parameter (space, energy, and static size of plot) (Figure 7). Setting the packet size and reading house temperatures were done through manager agents in the second phase. A run function calculation and the Dijkstra algorithm were executed in the third phase to determine the path for distributing energy between the first and last nodes. Finally, the result phase was presented by supplying energy to all devices or appearance of negotiation.

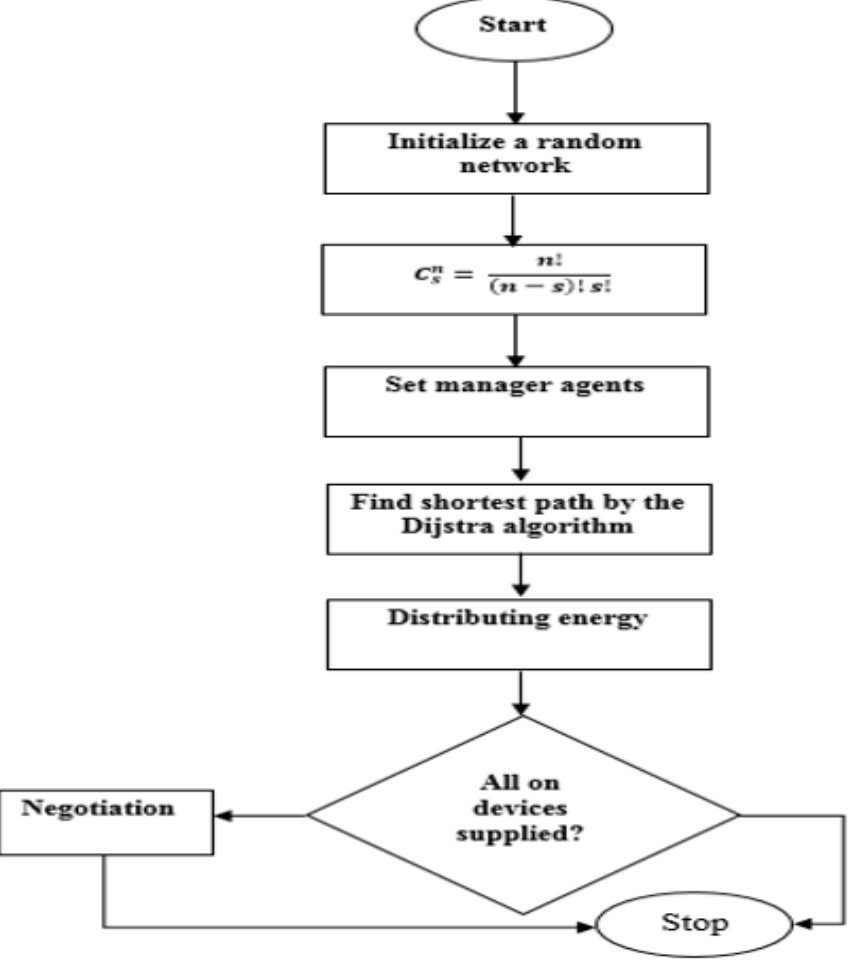

**Figure 7.** Proposed Distributed-Negotiated System Flowchart.

DNsys Architecture: the proposed system architecture included several houses, one mosque, and one hospital. Each one of these components considered a node with its data with related agents. Every agent had its objective, and it worked to achieve it. The system architecture components described are as follows:

- Ten houses (ten nodes), each one has a specific log for storing the temperature inside it alongside the measurement date and time. Changing temperature inside the house opens or closes the devices.
- One mosque, one node required full energy for cooling only during prayer times.
- One hospital that needs its power supply at full capacity with no discontinuity.
- Supplier device generates a random amount of energy to distribute among several nodes.
- Agent in the system performs the required operations and specifies the status for each one.
- Three manager agents are responsible for forming an agent coalition and the operations performed by their group, especially the negotiation process.
- A random network initializes each time the supplier is connected with the manager agents.

DNsys Topology: DNsys consists of heterogeneous agents, called functional agents, which supplement some energy functionalities (Figure 8). Agent communication language was used to communicate among agents that helped in understanding each other to initiate conversations.

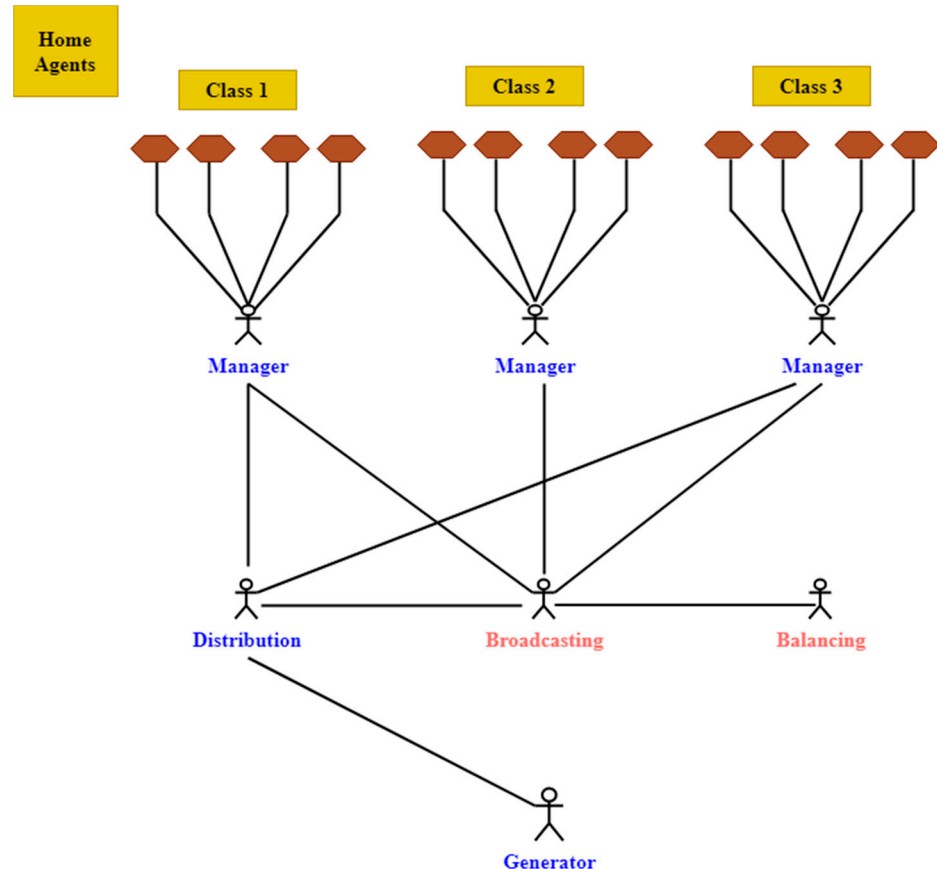

**Figure 8.** DNsys Hierarchy and Interaction between Agents.

The Generator Agent (GA) generates a random amount of energy for distribution. It distributes the energy to the manager agents of the system by finding the best way to separate the total energy supplied into the appropriate coalition. The Broadcasting Agent (BA) is responsible for sending messages to all agents in other systems via the network. BA communicates the system's energy value at the beginning of the distribution. GA sends a message containing the amount of additional power the system has to BA if it has sufficient power. BA then sends this amount to the balancing agent, which stores the energy in special batteries for later use. However, the negotiation begins when the system lacks sufficient power. Negotiation between the broadcasting manager and balancing agents occurs when MA receives a message from a house node containing the energy requirement and the

number of devices that require energy. BA broadcasts a message containing information from all other MAs, which then send messages to the balancing agent for negotiation. The balancing agent checks each group's energy needs and sends a broadcast message to all MA in the system with a request for energy. The request includes the number of devices to be turned OFF, the node it belongs to, and the amount of energy it can offer. The balancing agent communicates to the DA about the amount of energy which it can offer, and the coalition group name, if the request gets accepted. DA informs the balancing agent to send a verification message to all other coalitions to verify their energy sufficiency. Balancing agent then stores the nodes information which cooperated in the negotiation at the top of a list containing nodes that deserve energy. The process begins again if the requirement is refused. When energy is distributed, the balancing agent sends a message to MA to verify if the energy received was enough. At replying 'yes' from all MAs, the process immediately stops by sending a termination message to all agents every minute through BA. Whereas, when the reply is 'no,' BA checks to meet the required level of energy.

*Energy Consumption Calculations*

The system is designed for a compound constituting 12 nodes, represented by a particular agent with a specific number of cooling devices (air conditions). Table 1 presents the agent priority and condition of the system.

**Table 1.** Agent Priority.

| Agent | Priority | Class | Condition |
| --- | --- | --- | --- |
| Hospital agent | Highest priority | Upper class | 24 h/7 days a week |
| Mosque agent | Second priority | Moderate class | During prayer time |
| Houses agents | Based on classes | Lowest class | Device working |

Energy Consumption (EC) = Electric Ability (Kw) × Number of Homes (h).

A thermostat reduces the energy needs in a cooling device by 15% of the total consumed energy (1 ton = 3.5169 kw).

- Energy consumption for 1.5-ton device = 3.5169 × 1.5 = 5.3 Kw × 1 h = 5.3 kWh
- Energy consumption for 3-ton device = 10.6 Kw × 1 h = 10.6 kWh
- Energy consumption for 5-ton device = 17.6 Kw × 1 h = 17.6 kWh

## 5. Results and Discussion

In the study, the proposed CF mechanism (referred to as the modified approach) was compared with the basic coalition formation method without priority-based allocation and the remaining rate feature (referred to as the basic approach). Most of the data in the experiment were randomly created. They included resources for the distribution task, amount of energy, arrival time and leaving time of an agent, etc.

In a similar context, the present study compares the CF mechanism (referred to as the modified approach) with the basic coalition formation method without priority-based allocation, while retaining its rate feature (referred to as the basic approach). In this experiment, a substantial part of the data was randomly created, including distribution resources, amount of energy, arrival time, and leaving time of an agent. The experiment replicates the actual circumstances by randomizing this data. A random walk algorithm can be generated through a statistical distribution using a set of random numbers, which resemble the real attenuation factors as well as their proportion of changes to develop any attenuation factor such as distribution resources. These values, on average, indicated similar attributes to the ability of random models for accepting different random number generator seeds as input for allowing the same randomly generated data for replication when needed. Three experiments were performed to compare the modified approach (Distributed Negotiation system (DNsys)) and the basic approach (Non-Negotiable system algorithms (NNsys)). In the first experiment, agent-staying time differed for the purpose of

studying its effect on the coalition lifetime. In the second experiment, agent lifetime was extended to study its effect on the coalition lifetime. In the third experiment, the effect of agent staying time was measured based on the success rate of accomplishing a task.

### 5.1. First Experiment

An environment was created as a network with sufficient resources; however, the staying time of each agent in the network differed. An agent may stay in the environment for a long period, while another might stay for a shorter period. Figure 9 shows the staying time for the DNsys and NNsys. The number of agents was increased to increase the average staying time.

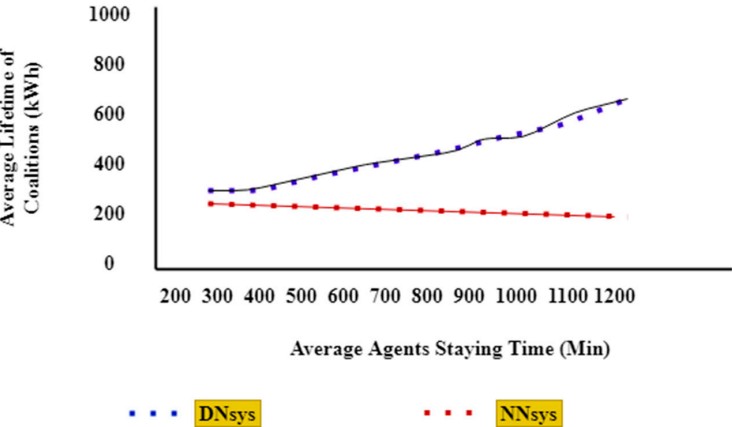

**Figure 9.** Coalition Average Lifetimes Verses Agents Average Staying Time of NNsys and DNsys in the First Experiment.

### 5.2. Second Experiment

It is observed that the agents' average staying time was in direct relation with the coalition lifetime, which was formed using the basic CF algorithm. For example, an increase in the agents' average staying time led to a slowdown in the increase in the coalition lifetime. Besides, whether agents were added into the environment or not, the average coalition lifetime hardly changed (Figure 10).

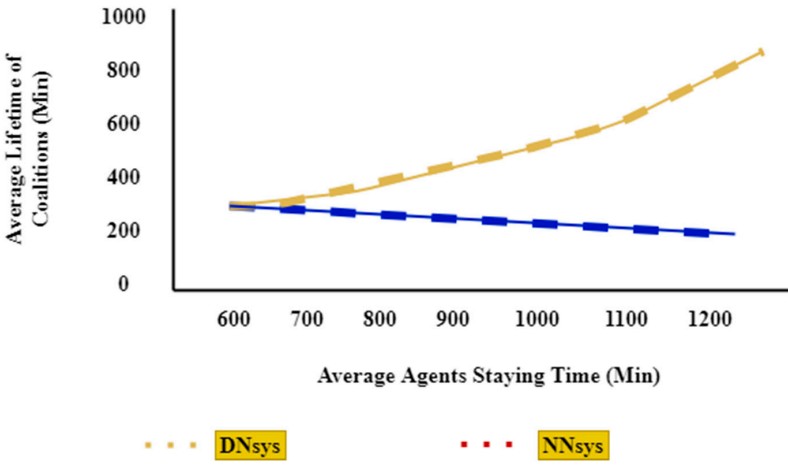

**Figure 10.** Coalition Average Lifetimes Versus Agents Average Staying Time of NNsys and DNsys in the Second Experiment.

### 5.3. Third Experiment

The third experiment dealt with distribution task accomplishment. Stable coalitions accomplished sufficient tasks despite the increase in the number of agents. The coalition lifetime increased when the priority-based allocation was used with the remaining rate feature that helped in the accomplishment of all the tasks with no loss of agents. Figure 11 shows that the success rate of the modified method was still more successful, i.e., 30 percent as compared to the basic approach.

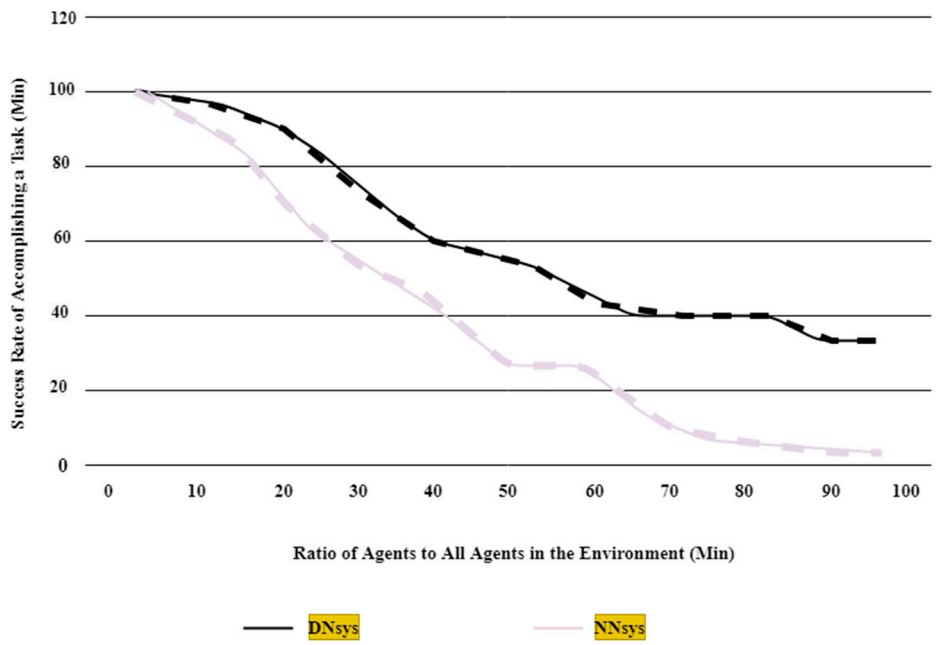

**Figure 11.** Experimental Results to Satisfy the Task Accomplishment of NNsys and DNsys.

### 5.4. Energy Consumption for Hospital

The compound's hospital agent is a highly integral and prioritized agent among the remaining agents. Table 2 presents energy consumption values for DNsys and the NNsys in minute-by-minute time intervals.

**Table 2.** Energy Consumption for DNsys and NNsys.

| Time Interval | DNsys (kWh) | NNsys |
|---|---|---|
| 1 | 340 | 346 |
| 2 | 346 | 346 |
| 3 | 330 | 346 |
| 4 | 325 | 346 |
| 5 | 335 | 346 |
| 6 | 324.8 | 346 |
| 7 | 330 | 346 |
| 8 | 340 | 346 |
| 9 | 345 | 346 |
| 10 | 337 | 346 |

As shown in Figure 12, the DNsys reduces energy consumption by approximately 3% as compared to the NNsys. In the NNsys, the energy consumed is 10,380 kWh during operation time while, the in DNsys, it is reduced to 10,059 kWh. Also, the DNsys consumed 96.9%, saving about 3% of the overall energy, while NNsys consumed almost 100% of the total energy.

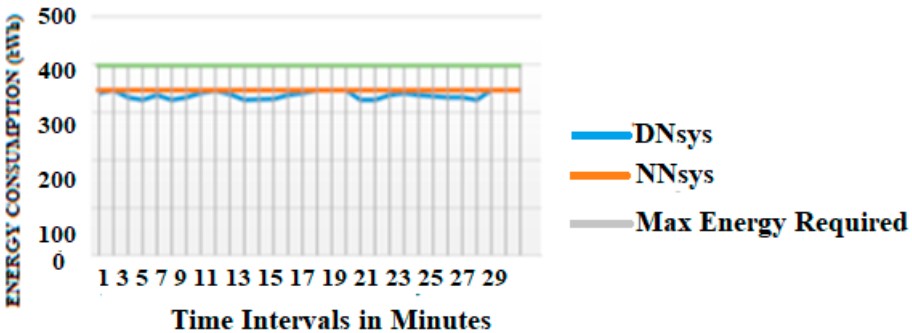

**Figure 12.** Hospital Agent Energy Consumption in DNsys and NNsys.

*5.5. Energy Consumption for Mosque*

The experiment was conducted from 10:00 a.m. until 2:00 p.m. As shown in Figure 13, the *x*-axis is the interval designated for Dhuhr prayer time, while the *y*-axis shows the energy consumption of when the system is ON or OFF. The system is OFF and consumes no energy until half an hour before each prayer time (11:30 a.m.). After it is turned ON, it consumes energy until the highest energy consumption is reached. Half an hour after prayer time, the units turned OFF, resulting in low energy consumption until the system turns off and no more energy is consumed. The DNsys reduced the energy by about 7% of the total energy consumed in the NNsys. During prayer time, the energy consumed in the NNsys was 742 kWh, while the DNsys consumed only 689 kWh. Therefore, the DNsys consumes only 93% and saves 7%, if a mosque generally consumes about 100% of whole energy.

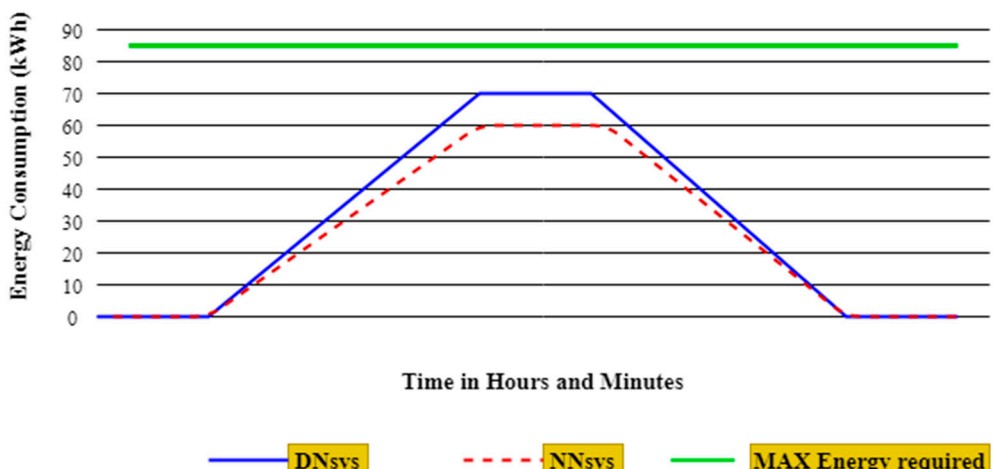

**Figure 13.** Energy Consumption of Mosque Agent in DNsys and Nnsysm.

Table 3 explains two scenarios that occur when the mosque device is ON. First, during prayer times, the mosque agent requires its energy and does not volunteer to other agents in the system. The second is when the mosque device is OFF, other house agents were left to wait until prayer time was completed. With the DNsys, the mosque agent checks its energy and starts to distribute power to other house agents once prayer time is over. There is no need for extra energy when a house agent does not consume more energy.

**Table 3.** Mosque Agent Volunteer.

| Agent | Mosque On | Mosque Off | Effect |
|---|---|---|---|
| House 1 | 26.5 | 26.5 | No effect |
| House 2 | 31.8 | 31.8 | No effect |
| House 3 | 37.1 | 37.1 | No effect |
| House 4 | 10.6 | 15.9 | Given power |
| House 5 | 0 | 10.6 | Given power |
| House 6 | 42.4 | 53 | Given power |
| House 7 | 31.8 | 31.8 | No effect |
| House 8 | 5.3 | 15.9 | Given power |
| House 9 | 5.3 | 15.9 | Given power |
| House 10 | 21.2 | 31.8 | Given power |
| Mosque | 74.2 | 0 | Volunteering to other agents |
| Hospital | 346 | 346 | Stable |

### 5.6. Energy Consumption for Upper-Class House

The energy consumption values of a house agent, using DNsys, are reduced by approximately 10.7% as compared to the NNsys. In the NNsys, the energy consumed is almost 1334.6 kWh during staying time, while the DNsys reduces it to 1191.5 kWh during the same time interval. As shown in Figure 13, the DNsys consumes only 89.27% of the energy, saving about 10.7% of the overall energy if NNsys consumes about 100% of the whole energy.

### 5.7. Overall DNsys Energy Consumption

Figure 14 presents the overall distributed energy in both the DNsys and NNsys. In the NNsys, energy is distributed to all agents based on the maximum distribution power, regardless of its use. This results in a large amount of energy loss. In DNsys, energy is distributed to agents only if a device needs energy. Therefore, DNsys benefits by using negotiation between agents.

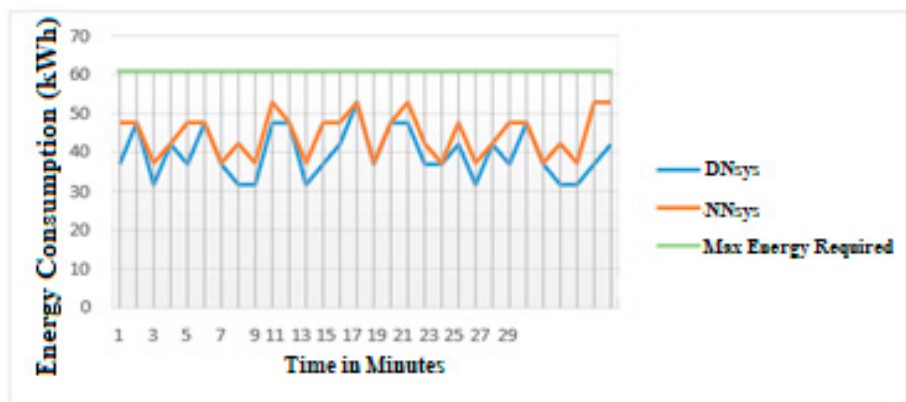

**Figure 14.** Energy Consumption of Upper-Class House.

The total energy consumption used in the NNsys was 950.2 kWh, while the proposed DNsys consumed a total of 632.2 kWh of energy Table 4 and Figure 15.

**Table 4.** Energy Distribution in DNsys and NNsys.

| Agent | NNsys | DNsys |
|---|---|---|
| House 1 | 53 | 26.5 |
| House 2 | 53 | 31.8 |
| House 3 | 53 | 37.1 |
| House 4 | 53 | 10.6 |
| House 5 | 53 | 0 |
| House 6 | 53 | 42.4 |
| House 7 | 53 | 31.8 |
| House 8 | 53 | 5.3 |
| House 9 | 53 | 5.3 |
| House 10 | 53 | 21.2 |
| Mosque | 74.2 | 74.2 |
| Hospital | 346 | 346 |
| TOTAL | 950.2 | 632.2 |

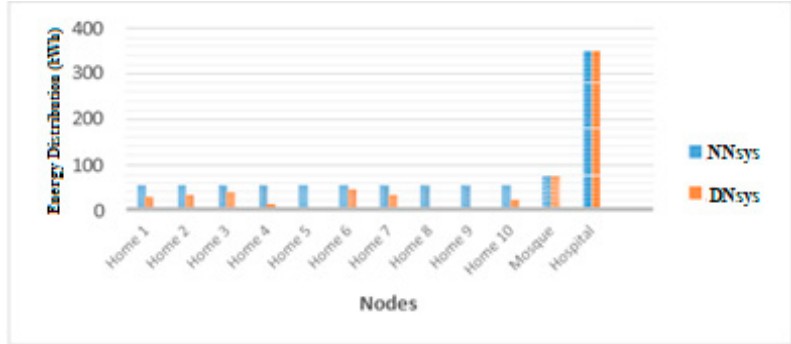

**Figure 15.** Energy Distribution in DNsys and NNsys.

### 6. Conclusions

The present study has mainly focused on coalitional value calculation in the coalition formation process, which includes distributional limitations. DCVC algorithm's basic version was used to distribute the coalitional value calculations among cooperative agents. DCVC was modified to reflect variation in the agent's computational speed. After developing the proposed algorithm, it was used to distribute the coalition value among cooperative agents by collecting relevant data from a residential area in Saudi Arabia. The performance of the proposed DNsys was calculated by comparing its results to NNsys. The results showed a reduction in cooling consumption by 20% after applying optimization algorithms. The amount of reduction in the cooling consumption reflects a 31% reduction in expected cooling costs, without affecting the household comfort. Therefore, the study concludes that DNsys provided better performance than the NNsys.

The need for alignment throughout wireless networks emerges naturally when enhancing the network efficiency. To this end, a number of practical conditions that motivate this study are the collection of information from the sink of wireless sensor networks. In fact, the use of the UAVs system, during the last decade, has been regularly elevated for cooperatively monitoring a predefined area with the use of a single drone. Small UAVs, in such systems, can remotely cooperate, take actions, and make decisions for fulfilling the objectives of a specific mission. Another essential application for WSN is called the distributed data storage, deployed in hostile environments. Sensors have to accumulate, in such application, and retrieve the sensed data until the visit of the mobile sink for collecting it. The proposed framework is also essential for a roadside base station in order to broadcast data to vehicles that can miss some packets because of their high-speed mobility. Lastly, the distributed surveillance camera network is monitoring moving targets into a predefined area with a number of cameras. All cameras are exchanging their own local information regarding each target for recovering all the conditions over the entire network.

**Author Contributions:** Conceptualization, A.A.M.; Formal analysis, M.M.A.L.; Funding acquisition, D.A. and D.A.; Methodology, A.A.M.; Project administration, A.A.M.; Validation, M.M.A.L.; Visualization, M.M.A.L.; Writing D.A.; All authors have read and agreed to the published version of the manuscript.

**Funding:** This work was funded by the Deanship of Scientific Research (DSR), King Abdulaziz University, under grant no. (9-15-1432-HiCi). The authors, therefore, acknowledge technical and financial support of KAU.

**Institutional Review Board Statement:** Not applicable.

**Informed Consent Statement:** Not applicable.

**Data Availability Statement:** The data will be available for review from the corresponding author, on request.

**Conflicts of Interest:** The authors declare no conflict of interest.

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
