# Peer review of "Coalition Formation among the Cooperative Agents for Efficient Energy Consumption"

_sustainability, doi:10.3390/su13158662_

Round 1
Reviewer 1 Report
The article presents an innovative, according to the authors, algorithm for distributing the value calculation among the cooperative agents, tested on the example of a selected part of Yanbu city.
After reading the manuscript, I get the impression that it is part of a larger study, rather carelessly skimmed in favor of this article. This is evidenced by, among others:
- well described with the DNsys system, but the lack of any mention of the principles of the NNsys system, which for a reader not oriented in teamatics is a difficulty in understanding,
- - experiments 1, 2 and 3 are described very vaguely and there is no clear explanation of why they were performed,
- - line 435 - in point 5.4 of Energy Consumption for Hospital there is a reference to, ... prayer time; ... "
- - line 465 - in section 5.6 of the Energy Consumption for Upper-Class House description there is a reference to, ... prayer time; ... "
- - data on energy consumption are presented in small pieces only for the Hospital
- - in the description of Energy Consumption for Mosque, point 5.5, there is table 3, which in my opinion is redundant at this point, it should be moved to another place, e.g. for a summary,
- - according to my opinion, the drawings do not contribute much to the article, e.g. on the basis of Fig. 15 it can be read that the houses 5, 8, 9 in the measured period did not consume energy in the DNsys system and hence the shown savings can be obtained ,
- - The conclusions are very laconic and do not show the authors' work as well as do not allow for understanding what was the idea of the article
Author Response
Comment 1: Page 14 –Figure 12 and Figure 13 and Figure 14 (page 15) Energy unit on the y-axis of Figure 12 (kW/h ????)
Comment 2: Page 14 – line 449: Where are the values (742kWh and 689kWh) in the graph in Figure 13?
Comment 3: Page 15 – line 465: 1,195.5kw/h ---> 1,195.5kWh
Comment 4: The article has a total number of references equal to 40. References are not up to date. Years 2021, 2020, 2017 - 0 references; Years 2019 - 3; 2018 - 1; 2016 -4. Only about 25% of the references are from the last 6 years.
|
Comment No. |
Page No. |
Line No. |
Actual Content |
Removed/ Edited/ Replaced Content |
Remarks |
|
1 |
- |
- |
- |
- |
Catered (Figure qualities are improved for Figure 12-14). |
|
2 |
- |
- |
- |
- |
Catered (Figure 13 is recreated) |
|
3 |
- |
- |
- |
- |
Catered (Unit is revised) |
|
4 |
4-5 |
182-206 |
- |
- |
Catered (New and recent citations are added) |
Reviewer 2 Report
Some small suggestions / corrections are highlighted in yellow in the text, including:
Page 14 –Figure 12 and Figure 13 and Figure 14 (page 15)
Energy unit on the y-axis of Figure 12 (kW/h ????)
Page 14 – line 449: Where are the values (742kWh and 689kWh) in the graph in Figure 13?
Page 15 – line 465: 1,195.5kw/h ---> 1,195.5kWh
Authors must correctly present the energy units (in the text and figures)
k is small and W is caps lock
kw / h - what does it mean?
The article have a total number of references equal to 40.
References are not up to date. Years 2021, 2020, 2017 - 0 references; Years 2019 - 3; 2018 - 1; 2016 -4. Only about 25% of the references are from the last 6 years.

Author Response

(The authors gave the same response as above.)

Reviewer 3 Report
This work has done a comprehensive survey on coalition development algorithm of distributing the coalition value among multiple agents. The findings are of interest to researchers in this field. I recommend it to be published after minor revision.
1. the abstract is not well written. the authors shall focus on the main findings.
2. Keywords could not reflect the key focuses of this paper.
3. please improve the front sizes and pixel of the figures to make them more readable.
4. In Section 5,5, Figure 13 shows no difference of energy consumption between DNsys and NNsys, which is not consistent with the discussions.
Author Response
Comment 1: The abstract is not well written. the authors shall focus on the main findings
Comment 2: Keywords could not reflect the key focuses of this paper
Comment 3: Please improve the front sizes and pixel of the figures to make them more readable
Comment 4: In Section 5,5, Figure 13 shows no difference of energy consumption between DNsys and NNsys, which is not consistent with the discussions
|
Comment No. |
Page No. |
Line No. |
Actual Content |
Removed/ Edited/ Replaced Content |
Remarks |
|
1 |
1 |
15-18 |
- |
- |
Catered (Findings are added in abstract) |
|
2 |
1 |
19-20 |
- |
- |
Catered (Keywords are added) |
|
3 |
- |
- |
- |
|
Catered (Font sizes and pixels of the figures are revised and improved) |
|
4 |
- |
- |
- |
|
Catered (Figure 13 is revised) |
Round 2
Reviewer 1 Report
Unfortunately, I still get the impression that the article is part of a larger part. Minor additions did not explain the points noted in the first review.
Author Response
Review Report Form
Open Review
(x) I would not like to sign my review report
( ) I would like to sign my review report
English language and style
( ) Extensive editing of English language and style required
( ) Moderate English changes required
( ) English language and style are fine/minor spell check required
(x) I don't feel qualified to judge about the English language and style
|
Yes |
Can be improved |
Must be improved |
Not applicable |
|
|
Is the content succinctly described and contextualized with respect to previous and present theoretical background and empirical research (if applicable) on the topic? |
(x) |
( ) |
( ) |
( ) |
|
Are the research design, questions, hypotheses and methods clearly stated? |
( ) |
(x) |
( ) |
( ) |
|
Are the arguments and discussion of findings coherent, balanced and compelling? |
( ) |
(x) |
( ) |
( ) |
|
For empirical research, are the results clearly presented? |
( ) |
(x) |
( ) |
( ) |
|
Is the article adequately referenced? |
( ) |
( ) |
( ) |
( ) |
|
Are the conclusions thoroughly supported by the results presented in the article or referenced in secondary literature? |
( ) |
( ) |
( ) |
( ) |
Comments and Suggestions for Authors
Unfortunately, I still get the impression that the article is part of a larger part. Minor additions did not explain the points noted in the first review.
All the above comments have been addressed in the latest version